# Robust Engineering for the Design of Resilient Manufacturing Systems

**Dimitris Mourtzis ***, **John Angelopoulos and Nikos Panopoulos**

Laboratory for Manufacturing Systems and Automation, Department of Mechanical Engineering and Aeronautics, University of Patras, 26504 Rio Patras, Greece; angelopoulos@lms.mech.upatras.gr (J.A.); panop@lms.mech.upatras.gr (N.P.)
* Correspondence: mourtzis@lms.mech.upatras.gr; Tel.: +30-2610-910-160

**Abstract:** As the industrial requirements change rapidly due to the drastic evolution of technology, the necessity of quickly investigating potential system alternatives towards a more efficient manufacturing system design arises more intensely than ever. Production system simulation has proven to be a powerful tool for designing and evaluating a manufacturing system due to its low cost, quick analysis, low risk and meaningful insight that it may provide, improving the understanding of the influence of each component. In this research work, the design and evaluation of a real manufacturing system using Discrete Event Simulation (DES), based on real data obtained from the copper industry is presented. The current production system is modelled, and the real production data are analyzed and connected. The impact identification of the individual parameters on the response of the system is accomplished towards the selection of the proper configurations for near-optimum outcome. Further to that, different simulation scenarios based on the Design of Experiments (DOE) are studied towards the optimization of the production, under predefined product analogies.

**Keywords:** discrete-event simulation; resilient manufacturing design; design of experiments



## 1. Introduction

Manufacturing is an extremely important sector of the global economy, accounting for 15.391 percent of global GDP in 2019, representing an added value of US$ 13,779 trillion [1,2]. In particular, due to the rapid development of digital technologies, the manufacturing environment is changing rapidly. This rapid growth and adoption of emerging technology by manufacturing companies has created the so-called "Fourth Industrial Revolution" or "Industry 4.0. Industry 4.0 is associated with manufacturing trends and technologies such as cloud technology, cyber-physical systems, the Internet of Things, augmented and virtual reality, and many more [3,4]. Living in the Industry 4.0 era, where competitiveness is the dominant factor in the modern market landscape, the constant evolution of manufacturing systems constitutes an indispensable procedure. This ceaseless progress requires the investigation of various technologies and techniques in order to respond to the volatile changes in the demands of the customers. Given the plethora of new technologies paired with numerous functionalities that have been integrated into manufacturing the last decades, it becomes apparent that the number of alternatives is increasing rapidly. Although searching for the optimum alternative can be proved strenuous, time-consuming and costly, state-of-the-art Information and Communications Technology (ICT) tools have enabled manufacturers to reduce development time, eliminate a significant part of the design and build cycles, as well as to address the need for more customer-oriented product variants [5]. In this context, a new trend regarding the efficient adaptation of modern manufacturing systems to external disruptions, is the so-called Resilient Manufacturing. Resilient Manufacturing is defined by the literature [6], as the ability of a manufacturing system to efficiently mitigate any external disruptions either derived from the supply chain of the company or resulted from the volatility of the market

demand. Further to that, the response of the system to these volatile changes must be as rapid as possible in order for the company to maintain their competitive advantage in the market landscape.

Towards that end, simulation comprises a focal point of digital manufacturing solutions since it allows the experimentation and validation of different products, processes and manufacturing system configurations [7]. Consequently, with simulation modeling the decision-making process can be facilitated as it provides useful insight regarding the behavior of the system under various conditions, in order to gain full perception of the system response under different, usually unpredicted scenarios. The number of combinations due to different settings can be countless; aiming to reduce the simulation effort, the main factors that affect the system are identified. On the contrary, engineers are constantly seeking ways for mitigating disruptions in manufacturing systems as fast as possible. Therefore, in many cases detailed simulation models and techniques are adequate since the computational domain is large and requires big computational resources and time. Consequently, robust engineering is a suitable candidate, especially in the case of resilient manufacturing, as it was described in the previous paragraph. More specifically, Pahdke in his book [8], defines robust engineering as a method for studying large numbers of decision variables with the bare minimum number of experiments required. In order to accomplish this, orthogonal arrays are utilized.

This research paper presents the modelling, the design and implementation of a copper tube industry by using DES based on real data derived from the existing manufacturing system. Ultimately the goal of this research work is to further increase the throughput of the manufacturing system based on the utilization of real data. Ultimately, the contribution of this research work is focused on the provision of a methodology for rapid adaptation of modern manufacturing systems to the volatile market demands, by minimizing the experimental runs to least minimum possible based on the utilization of Taguchi method and orthogonal arrays. Although the design of the framework is based on the modelling of a copper industry, the framework can be easily modified in order to accommodate other industrial models. On top of that, the ANOVA method is utilized in an attempt to highlight the effect of each decision variable on the industrial model. This facilitates the production engineer in identifying and strategically designing the manufacturing system, in order to meet the production goals. The rest of the paper is structured as follows. In Section 2 the most pertinent literature is examined in the field of the design of resilient manufacturing systems as well as the simulation technologies and techniques implemented under the Industry 4.0. Following, in Section 3 the line balancing problem is presented, and the simulation model of the current production line is developed. Then, the implemented methodology for the DOE is explicitly presented, based on the utilization of the Taguchi method. More specifically, the Taguchi method is used in an attempt to obtain a clearer direction in the experimental procedure so that the appropriate system settings, which satisfy the main target of the manufacturing design, can be easily recognized. The developed model is validated, using real data from the existing manufacturing system, and then the experimental setup along with the experiment results is presented in Section 4. Finally, in Section 5 conclusions are drawn and points for future development are also discussed.

## 2. State-of-the-Art

Simulation modelling and analysis is conducted in order to gain insight into these complex systems, testing new operating or resource policies and new concepts or systems before implementing them, as well as to gather information and knowledge without disturbing the actual system [9]. Simulation is a very helpful and valuable ICT tool in modern manufacturing. It provides decision-makers and engineers with a secure and low cost tool that provides fast analysis for the investigation of the system complexity as well as for identifying possible changes regarding the system configuration. Further to that with simulation, the operational policies which may affect the performance of the system or organization can also be examined minimizing intrusion, and thus cause disturbances to

the physical/actual system [10]. Simulation models are categorized into static, dynamic, continuous, discrete, deterministic, and stochastic. Static models comprise a system of equations, which represent the state of the system under investigation in a specific point in time [7]. If the relationships that compose the model are simple enough, it may be possible to use mathematical methods, such as algebra, calculus, or probability theory, to obtain exact information on questions of interest; this is called an analytic solution. However, most real-world systems are too complex to allow realistic models to be evaluated analytically, and these models must be studied with the utilization of suitable simulation models [11].

The traditional role of simulation is to present a "smart" business with a significant competitive advantage during the development, deployment and implementation of its plans and strategies. Simulations are accomplished by virtualizing processes with the use of tools, testing virtual models prior to application in the real world and proving to help: (i) performance prediction, such as latency, utilization and bottlenecks, (ii) disclosure of how the various components of a system interact, (iii) experimenting with and evaluating the merits of alternative scenarios, (iv) providing a knowledge base of the system configuration, (v) serving as a valuable means of demonstration and as a consequence of the above and finally, to (vi) promote decision-making [12]. Discrete event simulation (DES) as a discrete sequence of events is a form of computer-based modeling of a system. Simulation has a number of advantages over other operational research (OR) techniques, including the ability to experiment with any aspect of a business system [13]. The term "discrete" refers to the simulation progressing through time at mutually exclusive intervals. A mechanism is required to track the evolution of time, taking into account the dynamic nature of the systems to be modelled. This is accomplished using a tool known as a "simulation clock," which changes as events occur. As previously stated, DES models employ two prominent approaches as follows [11]: (i) Next Event Time Advances (NETA) and (ii) Fixed Increment Time Advances (FITA). Generally, NETA is more commonly used in simulation than FITA, because it is less complicated [12]. Finally, the effectiveness of DES and its flexibility result from its stochastic nature which makes it suitable for use in a wide range of applications, including warehouse operations, as a means of validating the performance of different indicators such as concerning material handling systems or order-picking and product location strategies [14]. Nonetheless, as computing processing power has increased, artificial intelligence (AI) has emerged, and IoT sensing has expanded in the modern warehouse, a new type of "simulation" paradigm known as the "Digital Twin" has emerged, allowing real-time control and digitization of a physical system [15–17].

Moreover, by modifying the production line for personalized production it is necessary to integrate Cyber Physical Production Systems to achieve the unique customer requirements and to achieve resilience [18]. Resilience is a quality that is related to robustness and prevents performance indicators from deteriorating. Under the resilient production control, unauthorized events that can cause bullwhip and ripple effects are managed. Five key functional requirements for handling events must be met by resilience: (1) selection of actions, (2) measurement of key performance indicators (KPI), (3) monitoring, (4) notification of fluctuations, and (5) adjustment. Resilience is achieved by meeting requirements 1, 2, and 5 in terms of production control [19].

This benefit perceived early from the researchers but also from the companies which in combination with the rapid evolution of computer systems beetled off exploiting the advanced decision-making assistant. The results of two simulation models to investigate the effects of push and pull systems on a printed circuit board manufacturing process at an electronics company in Ankara, Turkey presented in [20] while a diamond tool manufacturing system simulation is developed to predict the number of machines and the number of workers necessary to maintain desired levels of production by the same company [21]. As make-to-order (MTO) strategy is increasingly adopted by companies at the recent years, the management of the delivery dates of the orders have become a concern for researchers who attempted to address it by the means of simulation [22,23]. The complexity and the stability of manufacturing systems, introducing concepts based

on DES and nonlinear dynamics theory also investigated by [24]. Through DES models with the use of multiple conflicting user-defined criteria has been also used by [25] for examining the evaluation of the performance of automotive manufacturing networks under highly diversified product demand. Last but not least, regarding the design of lean manufacturing systems through simulation, a case study of an organization following a job shop production system to manufacture doors and windows is presented in [26]. It is quite discernible that simulation is preferred in many instances since it provides insight to numerous problems of manufacturing systems.

In order to comprehend the system response to the change of a parameter state, several experiments must be executed and still the conclusion is not sure that it will be valid. Experiments though can be proved costly and time-consuming. DOE begins with determining the objectives of an experiment and selecting the factors for the study. Well-chosen experimental designs maximize the amount of information that can be obtained from a given amount of experimental effort [27]. Towards that end, matrix experiments are preferred, since they provide a set of experiments where the settings of the system that need to be examined are selected, they change into various levels and then the data of all experiments are collected and analyzed. Following the data analysis, meaningful inferences for the effect of each parameter can be determined. Taguchi design method is a fractional factorial design which uses an orthogonal array that can greatly reduce the number of experiments [28]. Despite the fact that methods for designing experiments, addressed to simulation, have been investigated a long time [29,30], it is observed that are still adopted the simulation models that describe present-day production lines in order to provide a prompt inspection of the system factors. A two-stage sequential approach to design experiments for studying simulation systems which have additional stochastic constraints, so the input factor space is restricted and irregularly shaped suggested by [31]. A multi-objective formulation of the buffer allocation problem in unreliable production lines presented in [32] where the factorial design has been used to build a meta-model for estimating production rate, based on a detailed, DES model. In addition to that, the relationship between various factors leading to output yield strength of rebar has been investigated by the authors in [33] as also the decision process of evaluating and selecting shop floor improvement solution by integrating DOE in a DES environment [34] undoubtedly stated the usefulness of Taguchi method and DOE philosophy into simulation models of real production lines.

Although many studies addressed the manufacturing design using simulation, there is a lack of publications that combine framework application with the real data for validation and further experimentation. The results of individual modules often contradict each other because they refer to indirectly related manufacturing information and context which hinders the applicability of tools to real life manufacturing systems and pilot cases beyond the ones initially studied [35]. For this purpose, in the current research work, an existing line balancing problem in a copper tube production line is described and the simulation approach for decision-making support is presented, considering actual data derived from the workstations of the factory. Afterwards, the simulation model is developed and validated using the deterministic data from the industry, the data analysis in order to describe the model parameters with distributions is performed and finally the experimental design under the guidance of the Taguchi method. Following the target of the throughput maximization, the results of the experiments regarding the relative magnitude of each factor and the comparison with the real production data intend to empower the copper tube company to steer into decisions that will affect the production layout having as main goal the highest possible productivity.

## 3. Methods and Tools

This work presents the system modelling of a production line and the investigation of the criteria that will be able to lead efficient decision makings for increasing the production throughput while keeping the capacity utilization of the equipment high. The lack of

long-term insight of the production requirements and bottlenecks leads the company to poorly supported decision making, mostly based on tacit knowledge. Towards that end, at first place, the information collection regarding the whole copper tube factory is required. The working schedule of the production line working schedule is three shifts per day and seven days per week while the production amount is considered about 70,000 tons annually. The production accuracy is adequately high while the diversity of the production is also high, since more than 1000 different products can be produced. An important characteristic of each part is its linear weight (gr/m). The company products are presented below separated into two main categories: Level Wound Copper coil (LWC) and Pancake Copper tube (PNC). The products are used in various fields, such as plumbing, heating, ventilation, air-conditioning and refrigeration (HVAC&R), renewable energy applications and architecture, engineering, and industrial production.

### 3.1. Simulation Model

Referring to the actual production line, the procedure begins with the arrival of the raw material, namely the copper billet. The next step is the extrusion press where the billet subjects to 40,000 kN force. The extruded tubes are called mothertubes. Mother tubes undergo a series of dimensional reductions from this point and forward. For the final product, the number of demotions that are required is varying according to orders placed by the customers. Following, the production line is split into two routes where the first one refers to the drawbench and the second one to the cold pilger mill. In these two machines, the first demotions are performed. Next, the tubes are inserted into the spinner blocks. These machines are responsible for the higher percentage of the dimensional reductions in the production line. For example, an original tube diameter could be 35 mm and the final diameter 9.5 mm in or 6.35 mm. After the spinner blocks, the parts are directed to the final formatting machines in order to take the final form, for example, inner helix or coating, according to the order specifications.

As far as planning strategy is concerned, the MTO philosophy dominates. Although the manufacturer offers a defined variety of products, production profile is based on customers' orders. This strategy entails to limited buffers among the workstations and also a high dependency of the machines. Due to the high processing capabilities of the spinner block machines, specifically about 1200 m/min linear velocity and phase change less than one minute.

The manufacturing system, i.e., the copper tube production, has certain unique features that need to be considered during the modelling process. The manufacturer may offer a wide variety of different products, and specifically more than 1000 different products, which are inefficient to model, therefore they are omitted. It should be clear that not all parts of the production line are being simulated. Instead, only a certain portion of production line is modelled, in order to judge easily and quickly the response to various factors' change since indeed, a simulation model is supposed to be an abstraction and simplification of reality [36]. In collaboration with the production experts, the products were divided into eight groups (A–H) and the model of the production depicted in Figure 1 was created.

A simulation model of a complex system can only be an approximation to the actual system, so as to ensure that the model is not unnecessary complex [36]. Towards this end, an adapted production model was created in collaboration with the experts that simplifies the current production. The simulation model is validated to ensure the credibility of the results that will emerge from the experiments. The DOE in order to limit the simulation effort is executed and last but not least, experiments are conducted so useful conclusions can arise and assist the decision-making process towards the goal of production maximization. The flowchart of the simulation model is presented in Figure 2, while the development is taking place in Section 4. The five final machines, as they termed before, are referred as $M_i$ where $i$ ranges from 1 to 5.

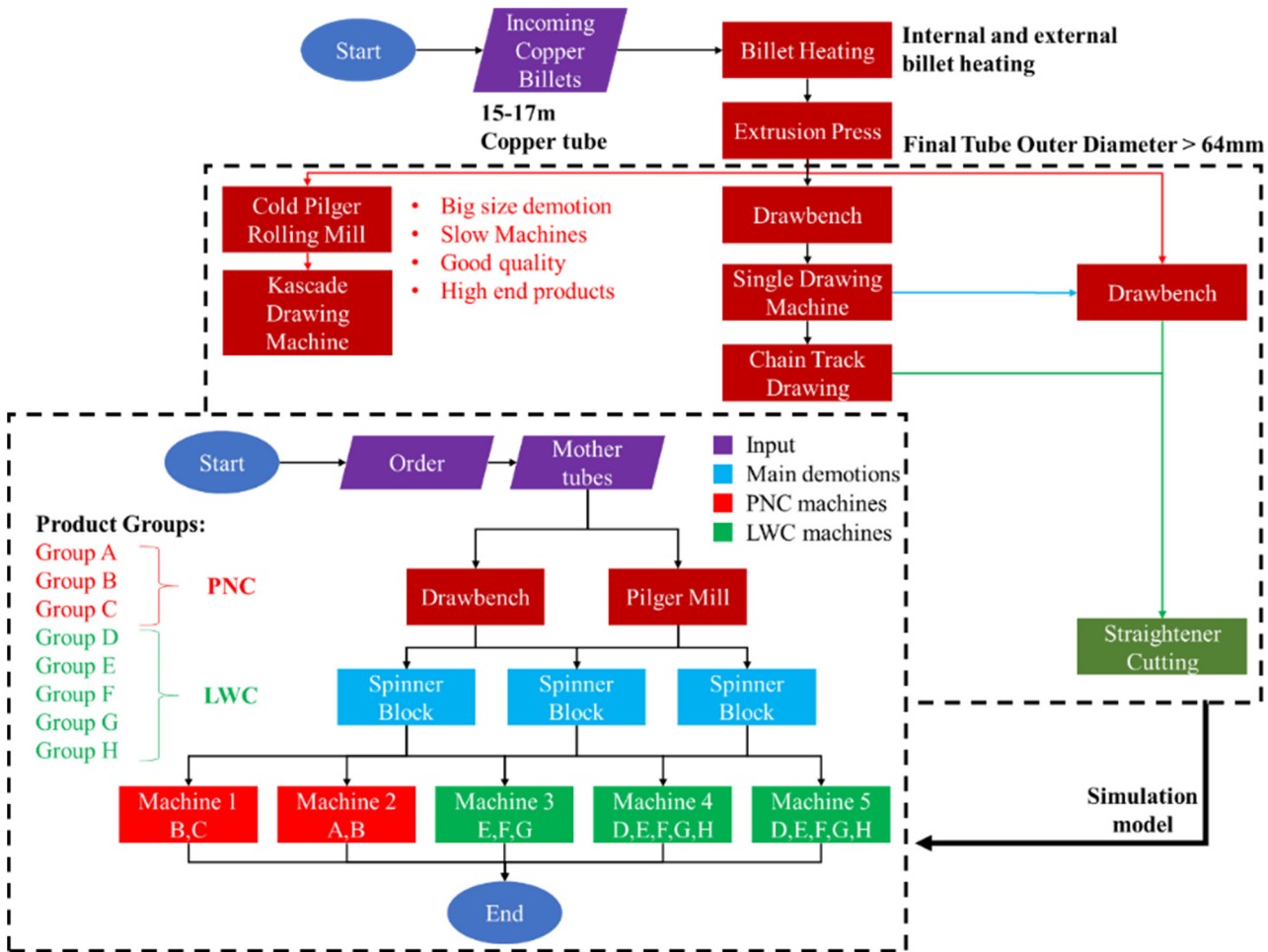

**Figure 1.** Line part simulation of the production.

### 3.2. Design of Experiments (DOE)—Taguchi Method

The procedure that has been followed for the experimental design has been divided into a series of steps which are explicitly presented in the following paragraph.

**Step 1:** Setting the objective of the experiment and selecting factors

In order to investigate the response of a system, the objective should be clearly defined. Subsequently, the parameters affecting the system should be determined. In the following paragraphs of this paper, the parameters are mentioned as factors. Many parameters usually affect the performance value characteristic. However, the most important factors have to be selected so that the analysis complexity will not be increased unreasonably. The parameters are realized as the control factors and are specified freely by the designer. Moreover, the designer is responsible for determining the best values of the control factors. On the other hand, noise factors cannot be controlled by the designer or their settings are difficult or expensive to control.

**Step 2:** Level identification for each factor

The multiple values that are assigned to each factor are called levels. After selecting the number of the factors, the levels of each one should be also determined. This step will be decisive for choosing the suitable Orthogonal Array (OA), which is discussed in the next step, Step 3.

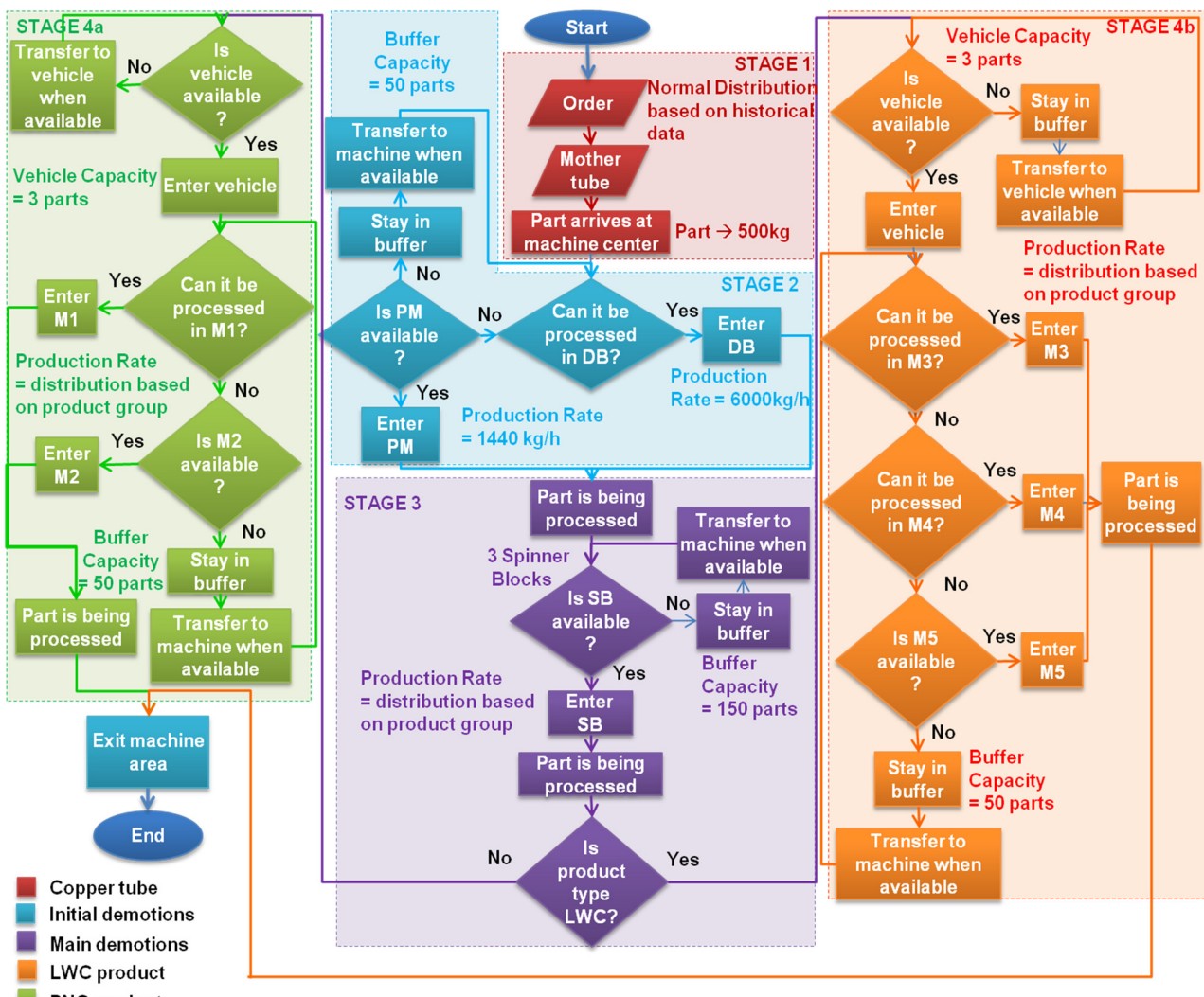

**Figure 2.** Flowchart of the proposed simulation model.

**Step 3:** Standard Orthogonal Array selection

The number of rows of the OA represents the number of experiments. In order for an array to be a viable choice, the number of rows must be at least equal to the degrees of freedom required for the case study. Similarly, the number of columns of the OA represents the maximum number of factors that can be studied using that array. Usually, it is expensive to conduct experiments, from a computing resources point of view. Therefore, the best practice is to utilize the smallest possible OA that meets the requirements of the case study [37]. The minimum number of experiments that are required for the implementation of the Taguchi method can be calculated based on the degrees of freedom approach, as per the Equation (1).

$$N_{Taguchi} = 1 + \sum_{i-1}^{n}(L_i - 1) \tag{1}$$

where,

$L_i$ stands for the level of each factor and $n$ for the number of factors.

**Step 4:** Selecting the appropriate objective function (Signal-to-Noise ratio)

The selection of the appropriate objective function for maximization in an engineering design problem is very important since it defines the value of the conclusions about the optimum levels. Usually, in a production line, the target is to maximize the overall

throughput. This type of problem is referred in the literature as the-larger-the better problem and the corresponding objective function is depicted below.

$$n = -10 \log_{10} \left( \frac{1}{n} \sum_{i=1}^{n} \frac{1}{y_i^2} \right) \tag{2}$$

where,

$n$ stands for the Signal-to-Noise ratio (S/N ratio)

$y_i$ is the response in each experiment $i$.

**Step 5:** Conducting the experiments

Provided that the control factors (A, B, C, D), the noise factors, the orthogonal array and the objective function have been defined, then the next step involves the execution of the experiments. For each experiment in the OA, the response should be measured and then the S/N ratio must be calculated. Noise factors should also have levels and each experiment will be conducted for different noise factors settings. Afterwards, all the data will be gathered, and the mean value is calculated which in turn represents the system response.

**Step 6:** Analysis of means

The variation caused by each factor to the overall mean is depicted in suitable graphs, where the overall mean is indicated and the average of each factor for every level is also depicted. In order to perform the analysis of means some critical values should be defined at first place. The overall mean represents the average of all the mean values calculated and presented in the main effects plot. For the calculation of the means, Equation (3) is utilized.

$$m = \frac{1}{n} \sum_{i=1}^{n} n_i \tag{3}$$

For each factor, the individual mean has to be defined. Therefore, the S/N ratios that resulted from the experiments that a factor participated in the particular level are summed up. For instance, with the use of Equation (4), the average of factor A at level 3 can be calculated.

$$m_{A3} = \frac{1}{3}(n_7 + n_8 + n_9) \tag{4}$$

Subsequently, the plots of factor effects are constructed where the deviation caused from every factor to the overall mean is depicted graphically.

**Step 7:** Analysis of variance (ANOVA)

Analysis of variance which practically is the decomposition of variance can provide a deeper understanding of the relative effect of different factors. The magnitude of each factor on the objective function η can be determined. However, attention should be given to the inferences that will arise from these calculations. The Taguchi method does not attempt to make any probability statements about the significance of a factor as commonly happens in statistics [37]. Consequently, for the analysis of variance basic values have to be calculated, by applying Equations (5)–(10).

$$SS = \sum_{i=1}^{n} (n_i - m)^2 \tag{5}$$

$$SS_A = 3(m_{A1} - m)^2 + 3(m_{A2} - m)^2 + 3(m_{A3} - m)^2 \tag{6}$$

$$MS_{factor} = \frac{Sum\ of\ squares\ due\ to\ each\ factor}{Degrees\ of\ freedom} \tag{7}$$

$$F_{ratio} = \frac{MS_{factor}}{Error\ mean\ square} \tag{8}$$

$$Variation\ \%\ due\ to\ factor\ A = \frac{Sum\ of\ squares\ due\ to\ factor\ A}{Total\ sum\ of\ squares} \tag{9}$$

$$\sigma_e^2 = \frac{sum\ of\ squares\ due\ to\ error}{degrees\ of\ freedom\ for\ error} \tag{10}$$

where $SS$ is the total sum of is squares, $SS\_A$ is the sum of squares due to factor A, $MS_{factor}$ is the mean square for each factor and $\sigma_e^2$ is the error variance. The Equation (6) refers to the factor $A$ but the formula is applied respectively for all the factors as also the Equations (7)–(9). $F$ ratio represents the variance ratio. A large value of $F$ means the effect of that factor is large compared to the error variance. It can be used to rank order factors. It should be mentioned that in robust design $F$ ratio is used only for the qualitative understanding of the relative factors.

### 3.3. Data Analysis

In order to assume the stochastic and statistical properties of the system, the analysis of the available data is required. Indeed, two types of dataset are recruited, converted into a utilizable form, and analyzed through statistical tools. The outcome of this procedure is the extraction of the distributions that can sufficiently describe the system.

### 3.3.1. Arrival Time Analysis

For the analysis of the arrival data, the software tool MATLAB [38] has been utilized. The first dataset consists of the arrivals. Each arrival refers to a certain product and includes the date and the number of products of the respective order. The strategy of the company is after receiving a specific order, determining the amount of the raw material that it is going to be used. The raw material is translated into number of 500 kg billets. Since every order was kept tracked in kilograms by the company accordingly, the kilos are converted into parts of 500 kg, as it happens in the real production line and they rounded up. For example, an order of 1300 kg will be transformed into 3 parts in the data analysis process. Similarly, a 6200 kg order will be transformed into 13 parts and so forth. The next step is the pre-process, which is focused on grouping the arrivals into weeks as indicated from the data format by the company. In fact, a new series is formed by displaying the sums of arrivals that arrived into the system during the same weeks. Afterwards, the attention is focused on finding a distribution that efficiently fits the final series. However, due to the fact that the number of observations is significantly low, namely only 56, no statistical analysis tool is valid for recruitment. In this context, normality is assumed without further investigation. From the process described above, it is defined that the distribution that best follows the amount of parts arriving at the system every 168 h, which in turn denotes that the interarrival time is assumed 168 has suggested from the company data. The results of the normal distribution approach regarding the mean value and the standard deviation are listed in the Table 1.

**Table 1.** Parameters of the Normal distributions for the arrival time of the parts.

| Group | Mean Value | Standard Deviation |
|-------|-----------|--------------------|
| A | 74 | 45 |
| B | 107 | 68 |
| C | 48 | 31 |
| D | 40 | 27 |
| E | 133 | 75 |
| F | 149 | 82 |
| G | 131 | 73 |
| H | 30 | 21 |

### 3.3.2. Processing Time Analysis

The second data consists of the number of orders per product group and the respective processing times and working machines. In this case, the processing time is the main item and the corresponding working machine helps link each processing time to a specific machine. The number of orders expresses how many times a specific product type is needed to be produced and as such, it can be utilized as frequency. In fact, a new dataset is created by attributing the process times to working machines and then displaying each processing time value as many times as its corresponding number of orders. The basic concept behind this procedure is to make processing times of the most frequently encountered orders to have the biggest statistical impact on the data.

Firstly, a visual tool is recruited by estimating the fitment of the distribution on the given data and plotting the empirical histogram along with the probability density function of each fit. The calculated parameters for the distributions are presented in Table 2.

**Table 2.** Processing time distributions regarding final machines.

| Group | Machine 1 | Machine 2 | Machine 3 | Machine 4 | Machine 5 |
|:---:|:---:|:---:|:---:|:---:|:---:|
| A | - | LogNormal (0.8074, 0.1975) | - | - | - |
| B | LogNormal (0.3839, 0.1414) | LogNormal (0.4745, 0.1609) | - | - | - |
| C | LogNormal (1.2084, 0.3170) | - | - | - | - |
| D | - | - | - | LogNormal (0.6767, 0.1754) | Gamma (24.268, 0.0286) |
| E | - | - | LogNormal (0.7328, 0.2304) | Gamma (22.2997, 0.036) | Gamma (16.267, 0.0519) |
| F | - | - | LogNormal (0.8481, 0.1279) | LogNormal (0.9091, 0.127) | Gamma (39.054, 0.0244) |
| G | - | - | LogNormal (1.0031, 0.1381) | Gamma (1.0507, 0.2198) | Gamma (19.65, 0.0628) |
| H | - | - | - | Normal (2.135, 0.465) | Normal (1.648, 0.136) |

Subsequently, the skewness estimation is recruited as a valuable indicator of the normality of the given dataset. In fact, according to [39], for a normal distribution, all moment of the odd level, except than the first one, are equal to 0. Apparently, the closer the skewness is to zero, the higher the probability of the system following normality. Subsequently, the recruitment of statistical hypothesis tests is executed that can estimate whether or not an actual signal can fit a predetermined distribution. In this context, Chi-square [40] and Cramer-von-Mises [39–41] are utilized. the next step is to follow a distribution fit procedure in order to have an accurate statistical description of the system.

Transitioning from theory to practice, a representative application of this statistical process is performed on Group A dataset. First, the empirical histogram is depicted in Figure 3. As shown, the best fits are attained through LogNormal and Gamma distributions.

As it becomes apparent from Figure 3, the Gamma and LogNormal distributions fit the dataset better, thus they approach the system more precisely, versus the other types of distributions.

The next step is to validate these initial assumptions by calculating some goodness-of-fit indicators, which are measures of how well the chosen distribution fits in the given dataset. First, the skewness is estimated as a first indicator of the normality of the signal and then the two goodness of fit Chi-square and Cramer Von Mises, hypothesis tests are recruited. Details on the tests are presented in Table 3. As shown, the LogNormal distribution is found to be the best fit for the given dataset. Similarly, the procedure is followed for all the group datasets.

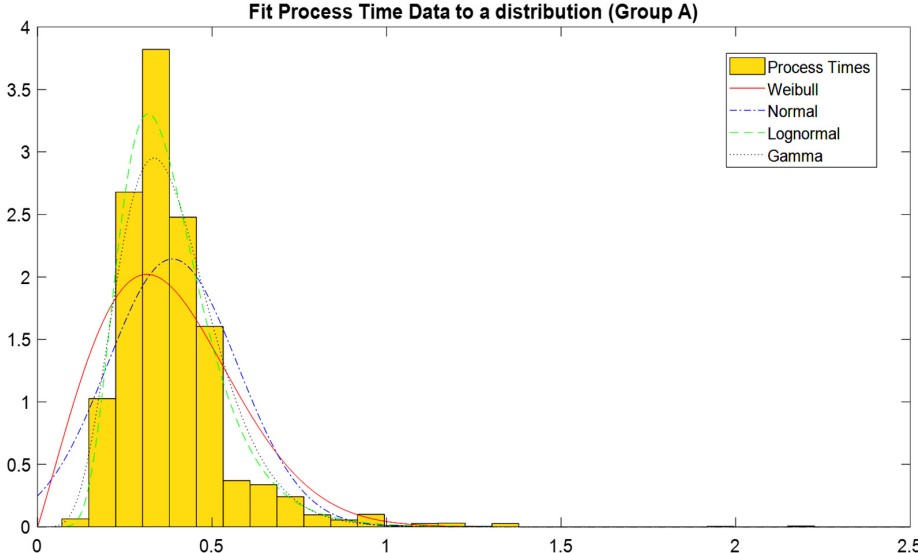

**Figure 3.** Empirical histogram along with Weibull, Normal, Gamma and LogNormal distribution fits.

**Table 3.** Details on the recruited statistical tests.

| Variables | Values | Interpretation |
|---|---|---|
| Skewness | 8.1784 | Normality Rejected |
| Cramer Von Mises Test ($p = 0.01$) | $H_0$: Accepted for LogNormal but rejected for Gamma distribution | It can be assumed that LogNormal distribution describes best the process times |
| Chi-square Test ($p = 0.1$) | $H_0$: Accepted for both LogNormal and Gamma distribution | |

Summarizing the data analysis method that explained extensively is depicted graphically in Figure 4.

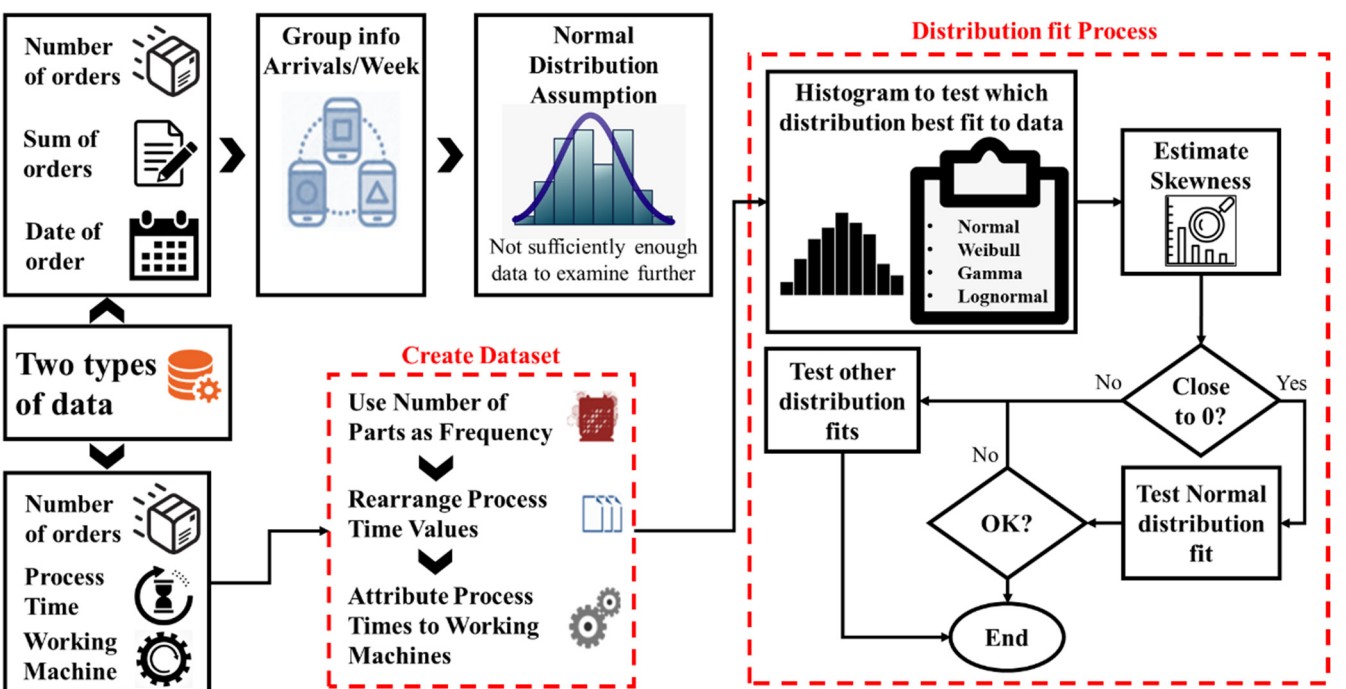

**Figure 4.** Flowchart for the data analysis process.

*3.4. Simulation Model*

The developed simulation model as well as the routing of the parts within the production line are presented and discussed in the following paragraphs. The simulation time is measured in hours, taking into consideration that the objective of the current study is the maximization of the annual throughput. The product groups and the physical product, which is the so-called mothertube, are considered as input to the model. The Machines for PNC products are $M_1$ and $M_2$ and for LWC products are $M_3$, $M_4$, and $M_5$. Initially, the physical part, which considered to be the copper billet of 500 kg, is represented by the part named "Mothertube" and appears on the model with interarrival time 0.067 h, namely 4 min. The second part for beginning the simulation is the part "A, B, C, D, E, F, G, H" and their interarrival time is 168 h, as indicated by the company, while the product mix is formulated through the pre-described datasets that are resulted form the real data since they represent the orders placed from the customers. Further to that, transportation times between some workstations were defined by the production engineers of the company.

The simulation model includes several entities which represent either a process or a piece of equipment/machine tool in the production line. each block shown in Figure 5 is referred and explained in detail below. Each "Mothertube" is combined with a demand part in the "Orders" block. The goal of this block is to combine the mothertube, which is produced under the "Push" approach with the corresponding order, in order to simulate the "Pull" approach hereafter. It is stressed out that the "Orders" block does not induce any time delays in the flow of the model. This technique is used in order to assign different characteristics to every product group such as processing time, transportation speed, and also for simulating the demand and the arrival of the physical product.

Each product group can be processed in particular machines and the routing is constraint by the standards set by the companies. Starting from the front stage of production, namely the drawbench ("DB") as well as the pilger mill ("PM"), the routing illustrated in Table 4.

**Table 4.** Routing regarding drawbench and pilger mill.

| Group | Drawbench | Pilger Mill |
|:-----:|:---------:|:-----------:|
| A | Yes | No |
| B | Yes | Yes |
| C | Yes | No |
| D | No | Yes |
| E | No | Yes |
| F | Yes | Yes |
| G | Yes | Yes |
| H | Yes | No |

Whenever a product group can be processed by both machines in order to decide where it should be stored, it is given priority to the buffer that has a smaller number of parts so there will be a balance between the processing parts in each machine for avoiding the congestion and utilizing properly the equipment.

The routing stage is followed by the drawbench and the cold pilger mill, which in the simulation model are represented by the "DB" and "PM" machines respectively. Their production rate is derived from the same normal distribution for all the products at this level. The value of processing time is 0.083 h and 0.347 h per part respectively. Following the spinner blocks, machines that are responsible for the main demotions in the product are represented as "Spinner Block". Three identical spinner blocks have been included while their processing time is following the normal distribution, according to the data analysis and it varies among the groups. the final stage in the simulation model involves the five machines for shaping the final product and their graphics are depicted as $M_1$, $M_2$, $M_3$,

$M_4$, and $M_5$ in Figure 5. The two latter machines, i.e., $M_4$ and $M_5$ are assigned to process only PNC products as long as the rest are processing explicitly LWC and their cycle time is following the LogNormal, normal, or gamma distribution based on the product group. Finally, there are separate buffers per product group for storing the completed parts.

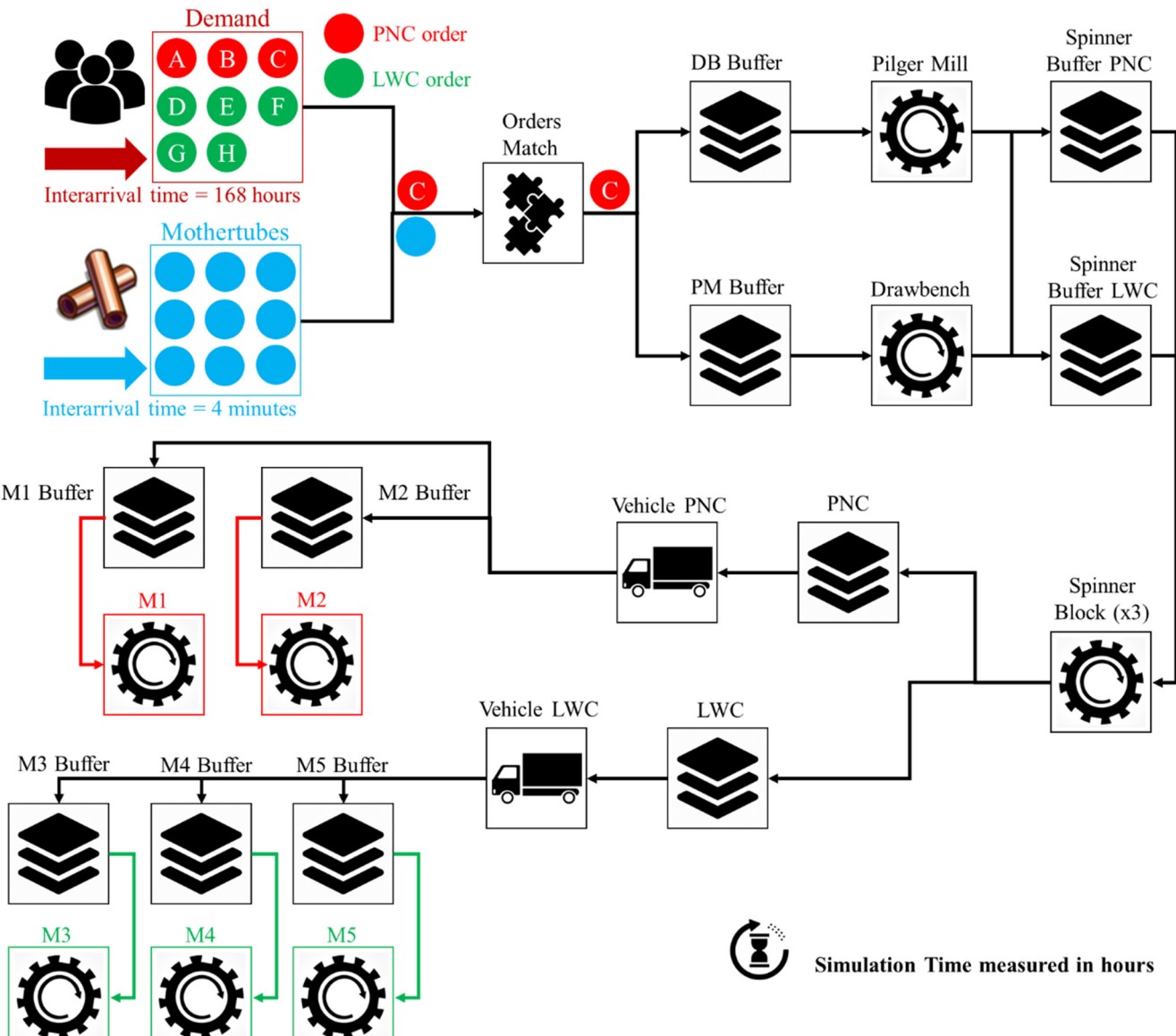

**Figure 5.** Structure of the simulation model.

As mentioned in the previous paragraph, the company follows a MTO strategy which denotes that low capacity buffers exist between the workstations. Taking into consideration that the WITNESS software requires that each machine is required to have a buffer before, low-capacity buffers are included in the model. Buffers for the demand and mothertubes are placed at the start of the production line. the capacity of the first buffer is unlimited while the capacity of the second buffer is set to 50 parts. The next buffers, i.e., the "DBBuffer" and the "PMBuffer", are located before the drawbench and the cold pilger mill machines. Concretely, each of these buffers can hold up to 50 parts. The reason for including two separate buffers is just for facilitating the routing process, due to the fact that not all products can be processed in both machines. Lastly, two buffers have been placed before the spinner blocks, namely the "Spinner Buffer LWC" and the "Spinner Buffer PNC", which

are capable of storing up to 50 parts each one and likewise are individually storing PNC and LWC in order to facilitate the part routing process.

Last but not least, as there is transportation time between the spinner blocks and the final machines, vehicles that are supported by tracks introduced purposefully to the model in order to simulate this delay. Consequently, four (4) tracks and two (2) vehicles are implemented in the model. The speeds of the tracks have been defined according to the indications of the production engineer, which were discussed in the previous paragraphs.

### 3.5. Design of Experiments Based on Taguchi Method

In this section, the application of the Taguchi method on the current model is going to be presented. after developing the simulation model, the next target is to limit the number of experiments that will be performed to the absolute minimum required, and for this purpose, the procedure described in the previous section is executed. Consequently, meaningful results for the factor that affects the most the performance value have been extracted. The objective of the presented use case is to maximize the production throughput. After studying the problem, discussing with the production engineers of the copper tube industry, the simulation model developed and then the factors selected. the key factors affecting the throughput as well as their corresponding levels, are listed below:

- Interarrival time (A)
- Buffer Capacity (B)
- Number of final machines (C)
- Number of Spinner Blocks (D)

The number of final machines (C) refers to the machines that are in the final stage of the production line, i.e., $M_i \forall i \in [1, 5]$. Certainly, the factors that influence the throughput are not only the above-mentioned. Many factors have an impact on the throughput, such as the transportation time, the number of the drawbench and pilger mill machines, but those that have the higher magnitude of the production results have been selected. For each factor, three levels are selected. In Table 5 the correlation between the Taguchi method factors and the key parameters of the simulation model is presented.

**Table 5.** Factor levels setup.

| | Levels | | | Unit |
|---|---|---|---|---|
| Factors | 1 | 2 | 3 | |
| Interarrival Time (A) | 96 | 168 | 240 | Hours |
| Buffer Capacity (B) | 20 | 40 | 60 | Parts |
| Number of final machines (C) | 4 | 5 | 6 | |
| Number of Spinner Blocks (D) | 2 | 3 | 4 | |

## 4. Results

As discussed in the previous sections, the developed simulation model can be approached with four factors and three levels for each one. For the calculation of the degrees of freedom, Equation (1) is used. The number of the degrees of freedom is nine (9). After this procedure, it is possible to choose the best fitted orthogonal array. If no interactions are taken into consideration, the suitable orthogonal array is the L9. The degrees of freedom should be equal or less than the rows of the orthogonal array.

In Table 6 the order that the experiments will be conducted in order to define $n$ is displayed. The observation $n$ corresponds to the S/N ratio as mentioned in Section 3. Therefore, after performing every experiment in the row as many times as the designer appraises, the mean value of all the replications is the $y$ that fills the objective function and $n$ is the calculated signal-to-noise ratio. Since it is desired to maximize the production throughput, the objective function has to selected according to this criterion. Consequently,

bearing in mind that the type of problem is the-larger-the-better, the objective function that is selected is:

$$n = -10 \log_{10} \left( \frac{1}{n} \sum_{i=1}^{n} \frac{1}{y_i^2} \right) \tag{11}$$

where $y$ is the response in experiment $i$.

**Table 6.** The resulting L9 array for the use case.

| Experiments | Interarrival Time (A) | Buffer Capacity (B) | Number of Final Machines (C) | Number of Spinner Blocks (D) | Observation (*n*) |
|---|---|---|---|---|---|
| 1 | 96 h | 20 parts | 5 | 2 | *n*1 |
| 2 | 96 h | 40 parts | 6 | 3 | *n*2 |
| 3 | 96 h | 60 parts | 7 | 4 | *n*3 |
| 4 | 168 h | 20 parts | 6 | 4 | *n*4 |
| 5 | 168 h | 40 parts | 7 | 2 | *n*5 |
| 6 | 168 h | 60 parts | 5 | 3 | *n*6 |
| 7 | 240 h | 20 parts | 7 | 3 | *n*7 |
| 8 | 240 h | 40 parts | 5 | 4 | *n*8 |
| 9 | 240 h | 60 parts | 6 | 2 | *n*9 |

Following the selection of all the appropriate values, i.e., factors, levels, etc., the experiments that are defined in the orthogonal array with the respective settings are executed. In this case, 90 replications are performed for each experiment. "Demand" is set as the noise factor of the system. In the current use case, the demand follows a normal distribution. As such, three normal distribution with different parameters are selected. Further to that, "Demand" can be realized as the combination of the demand for each product group. Consequently, the product mix is formed. The normal distribution for the Demand in level 2 results from the real data provided by the copper tube industry. The other two levels of the noise factor have been set and the experiments can be conducted. Concretely, 30 replications for each experiment. In the following Table 7, the mean values for each experiment iteration as well as the corresponding average value is presented. For the analysis of the results, Excel and MINITAB have been utilized. With the use of the objective function, the average of the throughput is calculated. Then, by substitution of the calculated value of the average in y, the S/N ratio, i.e., *n*, has also been calculated.

**Table 7.** Results from the 9 experiments and S/N ratios.

| Demand Level#1 | Demand Level#2 | Demand Level #3 | Average of the Mean Values | Observation | S/N Ratio |
|---|---|---|---|---|---|
| 29,010.53 | 26,836.96 | 23,882.36 | 26,576.62 | *n*1 | 88.48 |
| 33,917.00 | 39,044.30 | 32,934.73 | 35,298.67 | *n*2 | 90.95 |
| 33,950.83 | 49,171.70 | 40,064.23 | 41,062.25 | *n*3 | 92.26 |
| 19,301.30 | 34,869.93 | 28,833.13 | 27,668.12 | *n*4 | 88.83 |
| 19,308.50 | 29,127.03 | 26,859.66 | 25,098.40 | *n*5 | 87.99 |

Since the S/N ratios are available the analysis of means is possible to take place. As indicated in the methodology section for each factor in each level the mean is calculated as it is referred in (4). The results from this process have been compiled in Table 8. At the end of this table, in the row "Rank" it can be recognized the importance of every factor. This rank is based on the comparison between the higher and the lower S/N ratio that has resulted from for each factor. So, if we observe the factor "Interarrival", it is obvious

that between level 1 and 3 the difference is greater than in the other factors. So, this is the reason that is placed in the rank 1.

**Table 8.** Average *n* for each factor level.

| Larger Is Better | Interarrival Time | Buffer Capacity | Number of Final Machines | Numbers of Spinner Blocks |
| :---: | :---: | :---: | :---: | :---: |
| | (A) | (B) | (C) | (D) |
| 1 | 90.57 | 88.27 | 88.63 | 87.89 |
| 2 | 88.87 | 88.86 | 89 | 89.4 |
| 3 | 87.43 | 89.75 | 89.24 | 89.58 |
| Rank | 1 | 3 | 4 | 2 |

Substantially, each cell represents the variance that causes the particular factor level to the overall mean. Then, the plot of the factor effects is shown in Figure 6.

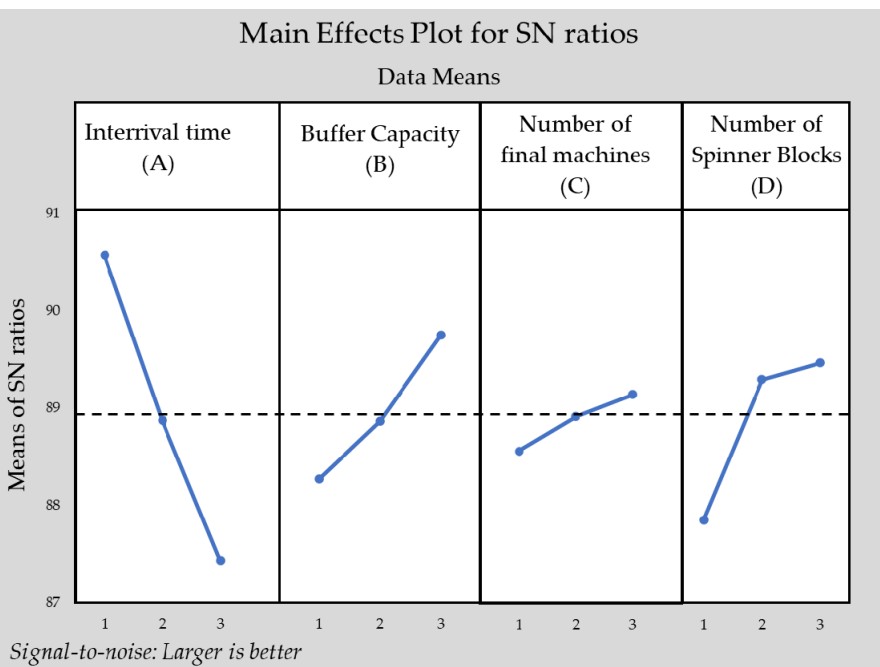

**Figure 6.** Plot of factor effects.

The dashed line in the middle of the diagram refers to the overall mean of the analysis. In this way it is apparent the deviation that every factor level causes to the overall mean. The final step in which the magnitude of the selected factors will become clearer. In order to perform the analysis of variance some values should be predefined. The total sum of squares is calculated using Equation (6), which is presented in Table 9 and the sum of squares due to each factor is depicted.

**Table 9.** Average of *n* and sum of squares due to each factor.

| Levels | A | B | C | D |
| :---: | :---: | :---: | :---: | :---: |
| 1 | 7.80046875 | 1.41796875 | 0.3217688 | 3.41866875 |
| 2 | 0.02296875 | 0.02851875 | 0.0054188 | 0.58741875 |
| 3 | 6.99976875 | 1.88416875 | 0.2394187 | 1.16251875 |
| Sum of squares due to each factor | 14.82320625 | 3.33065625 | 0.5666063 | 5.16860625 |

In the interest of gaining the most information from a matrix experiment, all or most of the columns should be used to study the desired parameters. As a result, no degrees of freedom may be left to estimate error variance [9]. The analysis of variance supposing zero error term is executed using Equation (9) as it is apparent in Table 10.

**Table 10.** ANOVA table excluding error.

|  | Degrees of Freedom | Sum of Squares | Mean Square | Contribution Percentage |
|---|---|---|---|---|
| Interarrival Time | 2 | 14.8232062 | 7.41160312 | 62.10347579 |
| Buffer Capacity | 2 | 3.33065625 | 1.66532812 | 13.95415582 |
| Number of final machines | 2 | 0.56660625 | 0.28330312 | 2.373860077 |
| Number of Spinner Blocks | 2 | 5.16860625 | 2.58430312 | 21.65445233 |
| Error | 0 | 0 |  |  |
| Total | 8 | 23.8685614 |  |  |

The contribution percentages are depicted in Figure 7. The values above resulted from the formulas on the corresponding methodology section and the factor with the greater impact is the "interarrival" with 62.1% effect on the throughput as it was indicated by ranking in the analysis of means.

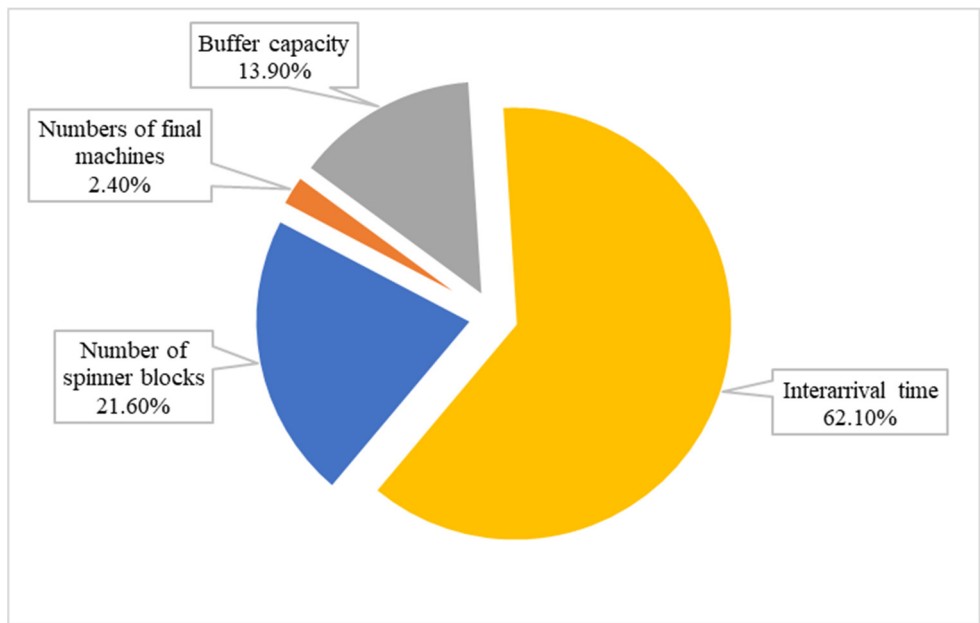

**Figure 7.** Contribution of each factor according to ANOVA table.

However, an approximate estimate of the error variance can be obtained by pooling the sum of squares corresponding to the factors having the lowest mean square. As a rule of thumb, the sum of squares corresponding to the bottom half of the factors corresponding to about half of the degrees of freedom be used to estimate the error mean square or error variance. In this case, the factors B, C are selected as they have the lower mean square value. This manner of error variance computation is called pooling. The results from the pooling operation are presented in Table 11.

**Table 11.** ANOVA table with pooled factors for error variance estimation.

| Factor | Degrees of Freedom | Sum of Squares | Mean Square | F (Ratio) | Contribution Percentages (%) | |
|---|---|---|---|---|---|---|
| Interarrival Time (A) | 2 | 14.823 | 7.412 | 26.16139 | 62.103 | |
| Buffer Capacity (B) | 2 | 3.3307 | 1.665 | | 13.954 | Pooled |
| Number of final machines (C) | 2 | 0.5666 | 0.283 | | 2.3739 | Pooled |
| Number of Spinner Blocks (D) | 2 | 5.1686 | 2.584 | 9.1220 | 21.654 | |
| Error | 4 | 3.8973 | 0.974 | | 16.328 | |
| Total | 8 | 23.869 | | | | |

The calculation of error variance for this case is calculated as follows:

$$\sigma_e^2 = \frac{3.8973}{4} = 0.974315625$$

The inferences that arise from this procedure are that having control over the interarrival time has a great impact on the throughput. The number of spinner blocks is considered as a secondary factor which has an impact of about 21.6% on the throughput. It should be clarified that this is not an analysis that provides accurate results of the system, especially when it changes dynamically, but offers to the designer the capability of identifying the main contributors in the performance value and have a prompt inspection on the system.

Towards the purpose of validating the simulation model, so the results can be credible and measurable, at first place, the deterministic data from the company inserted into the model and then the comparison between the deterministic and the stochastic model is carried out. The comparison is performed between the two models in order to find the error inserted when assuming a normal distribution for describing the arrival times. The model was simulated for one-year runtime, which is equal to approximately 8600 h.

As it has already been stated, the aim of this research work is to further increase the throughput of the manufacturing system based on the utilization of real data. To that end, calculations on the Signal to Noise ratio by posing as a criterion the bigger the better in order to achieve the highest total parts production were done. After that, experiments were conducted for What-If scenarios and the output was maximized (44,000 coils). Moving on, we have used the advanced experimenter mode of the simulation tool and experiments were done for different values of the interarrival time for the two main product categories. The purpose was to find the best combination of interarrival times. Moreover, this was done to achieve high utilization of the machines, maximum number of produced parts and low Work-In-Process (WIP) at the same time. Finally, with the implementation of this method, the cooper industry managed to produce 3000 more products in comparison with the current production model."

Multiple runs have been performed for the evaluation of the proposed methodology. The overall result for the evaluation is taken as the mean of the results for each run. Increasing the number of runs per evaluation increases the time taken to perform the experiments but it reflects better the variability that might be present in the results (Table 12).

**Table 12.** Deviation percentage between deterministic and stochastic model.

| | Arrival Data with Normal Distribution | Deterministic Arrival Data | Difference (%) |
|---|---|---|---|
| **Weight of products (*t*)** | 19,176.72 | 19,815.065 | 3.22 |

The percentage of the deviation of the total produced parts between the deterministic and the stochastic model, is 3.22% (Table 12) which is considered valid in the presented case by the engineers of the company in cooperation with the simulation team, as there is no such

thing as absolute model validity, nor is even desired [36]. A simulation model is always trying to obtain some meaningful inferences and is not to provide precise results acquired from certain calculations. As a result, every simulation project is a unique challenge to the model development team and the credibility of the model can be based on the evaluation from the team [42].

Another comparison between the deterministic and the stochastic model against the real number of products is performed in order to perceive the error that generated from the model assumptions. The total output weight is 20,355 tons. For the final validation of the model the differences between the total weight of the orders that arrived at the system and the total produced weight for both deterministic and stochastic model are presented at the second column of the Table 13. In the last column, the difference between the weight of the real orders arrived, i.e., 20,355 tons, and the total orders arrived assuming a normal distribution is presented.

**Table 13.** Comparison of the real number of products.

|  | Weight of Produced Parts (*t*) | Difference (Demand-Production) | Weight of Orders Received (t) | Difference (Real Demand-Distribution Demand) |
|---|---|---|---|---|
| Deterministic model | 19,815.1 | 2.65% | 20,354,6 | 0.002% |
| Stochastic model | 19,176.7 | 5.78% | 19,563,3 | 3.9% |

Presented in Table 13, the diversion simulation model which includes normal distribution for the arrival times of each product group is calculated around 5.78% less than the real amount of demand. Consequently, taking into consideration the abovementioned calculations, the validation of the model shows that it may be used to simulate the actual production.

## 5. Conclusions and Future Work

Summarizing, this work discusses the simulation modelling of a copper tube production line utilizing actual industrial data with the purpose of identifying the contribution of the individual major factors. Due to the countless configurations of the system, the factor with the higher impact is crucial to be defined in order to reduce the simulation effort towards the selection of the optimal settings according to requirements set by the company. The DOE methodology according to Taguchi presented and applied for perceiving qualitatively the magnitude of the most significant factors to the response of the system. Finally, after determining the most important parameter, a set of experiments were conducted and resulted to the conclusion that the interarrival time of products should be balanced in a manner, so that the system is capable of yielding the maximum throughput while properly utilizing the available resources, i.e., the machine tools. As the industry moves towards the establishment of the simulation modelling and the transition of traditional structural industry into the digitalized model where a plethora of scenarios can be tested, further work can be focused on elaborating more historical data regarding the demand so the distribution approach will be more reliable. Additionally, the seasonality of the demand must be analyzed and modelled properly. Furthermore, the equipment breakdown is possible to be included in the model after its proper modelling through historical data analysis. Generally, the opportunities to use simulation for facilitating the decision-making processes during normal operation of the system either address to maintenance either to a possible investment by simulation-based forecasting has already become a trend and surely requires further investigation in the existing manufacturing lines.

**Author Contributions:** All authors have participated in the modelling of the research project. More specifically D.M., supervisor, J.A., research, conceptualization, software, and original draft preparation, N.P. investigation, writing, original draft preparation. All authors have read and agreed to the published version of the manuscript.

**Funding:** This research received no external funding.

**Institutional Review Board Statement:** Not acceptable.

**Informed Consent Statement:** Not acceptable.

**Data Availability Statement:** Not acceptable.

**Conflicts of Interest:** The authors declare no conflict of interest.

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
