# Peer review of "Robust Engineering for the Design of Resilient Manufacturing Systems"

_applsci, doi:10.3390/app11073067_

Round 1

Reviewer 1 Report

This manuscript is hastily written as it fails to highlight the original research of the paper. Perhaps, after a thorough overhaul, it may be possible to raise the quality and contents of this manuscript to the publishable one on the journal "Applied Sciences." However, the manuscript suffers from another serious issue; namely, it is all about "copper manufacturing." This reviewer is wondering why not submit this paper to a journal specializing in mining &  minerals, as this manuscript is overwhelming about a case study on copper production. 

This paper consists of six sections. Section-1 "Introduction" is shallow and lack any useful information. In this section, a reader would at least expect definitions for "Robust Engineering" and "Resilient Manufacturing." Even though the tile emphasizes these two terms, there is no definitions or explanations for these terms. 
Section-2 "State-of-the-Art" is also shallow as it describes only simulations. The authors should at least establish the link between simulations (or digital twins) and "Robust Engineering"/"Resilient Manufacturing."    
Section-3 "Proposed Framework" presents a simulation model of copper manufacturing and a 7-step statistical approach for analyzing data from the simulation model. The 7-step statistical approach is named "the design of experiments (DOE) methodology".
Section-4 "Production simulation model based on the DOE" presents the simulation model and the details.  Section-5 "Experiments" presents some of the parameters that were varied. Section-6 "Results and discussion" presents the simulation results and the analysis of the results.   

The reviewer's conclusion:
The manuscript presents some copper manufacturing models and some ways of analyzing and experimenting with the simulation model. Thus, this manuscript is a candidate for submission to a journal on mining & mineral production. However, it is not clear why the paper is titled "Robust Engineering" or "Resilient Manufacturing". A lot more work is needed to establish this connection.

Author Response

Please check the attached file for the reply to the review comments.

Reviewer 2 Report

This work presents the simulation modelling of a copper tube production line utilizing actual industrial data with the purpose of identifying the contribution of the individual major factors.

Major strengths:

  1. Very nice presentation with excellent figuring and writing
  2. The idea is clearly presented
  3. The simulation results are given to support the proposed method
  4. References are nicely listed
  5. The proposed method is detailed presented from mathematical models to the actual simulation

Author Response

(The authors gave the same response as above.)

Reviewer 3 Report

  1. It would be better to follow IMRaD structure.
  2. It is not clear what was the methodology of your research. What were the questions to be answered? Were there overall hypotheses that your research was supposed to confirm/reject?
  3. How others can benefit from your results? It seems like a presentation of a modelling (however very important for studied organisation) for a specific case. How your findings are of interests of other researchers or practicioners?

Author Response

(The authors gave the same response as above.)

Reviewer 4 Report

  1. The paper use DES (discrete event simulation) and DOE (design of experiments) based on real data that obtained from a copper industry for designing and evaluating a resilient manufacturing system to enhance the throughput. The paper is generally well written but there are two errors of “Reference source not found” in Introduction section.
  2. In Section 3.2, the authors should explain the consideration of Taguchi method with ANOVA is required in the proposed DOE methodology in advance.
  3. The quantitative outcomes of the throughput enhancement should be highlight in discussion.

Author Response

(The authors gave the same response as above.)

Round 2

Reviewer 1 Report

(Nothing)